# Multi-Label Test-Time Adaptation with Bayesian Conditional Priors

**Qiru Li** [* 1]  **Ao Zhou** [* 1]  **Zhiwei Jiang** [1]  **Zifeng Cheng** [1]  **Cong Wang** [1]  **Yafeng Yin** [1]  **Qing Gu** [1]

## Abstract

Multi-label recognition with frozen Vision-Language Models (VLMs) is brittle under distribution shift: standard zero-shot inference scores labels independently, ignoring co-occurrence structure and producing incoherent label sets where dominant concepts suppress weaker but compatible labels. We introduce Bayesian Conditional Priors (BCP) Estimation, a gradient-free test-time adaptation method that injects label dependency without tuning the backbone. BCP views zero-shot logits as a proxy for marginal posteriors under a fixed image-text likelihood and attributes shift-induced errors mainly to a mismatched label prior. For each test image, it selects a high-confidence *anchor* label and applies an anchor-conditioned Bayesian refinement. This update is closed-form in logit space and admits a pointwise mutual information (PMI) interpretation, explicitly promoting compatible labels and suppressing incompatible ones. BCP operates without target annotations by estimating anchor-conditioned priors online from the unlabeled test stream via lightweight second-order co-occurrence statistics, adding negligible overhead beyond a single forward pass. Across standard multi-label benchmarks and multiple CLIP backbones, BCP consistently outperforms strong TTA baselines, e.g., improving RN50 average mAP from 57.31 to 69.22 and ViT-B/16 from 62.61 to 71.79.

## 1. Introduction

Benefiting from pre-training on large-scale datasets, Vision-Language Models (VLMs) (Radford et al., 2021; Li et al., 2023; Zhang et al., 2024a) have demonstrated remarkable generalization capabilities (Du et al., 2023; Li et al., 2025).

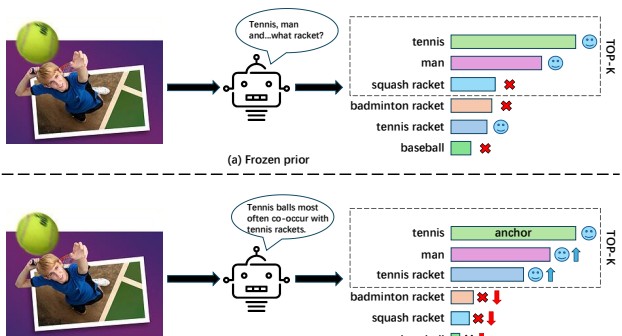

*Figure 1.* Comparisons between independent prediction and conditional calibration. While the Frozen Prior in (a) correctly identifies the dominant concept (*tennis*) but hallucinates visually confusing artifacts (*squash/badminton racket*), our Conditional Probability Prior in (b) corrects this by introducing an anchor-based dependency. This effectively aligns the top-$k$ predictions with the semantic context, ensuring that *tennis balls* co-occur with *tennis rackets*.

However, when confronting substantial discrepancies between training and testing distributions, representative models like CLIP still encounter severe challenges.

Test-time adaptation (TTA) (Shu et al., 2022; Karmanov et al., 2024) has emerged as a compelling paradigm for enhancing model robustness without target supervision. However, the exploration of TTA within multi-label recognition scenarios remains notably limited. On one hand, the naive extrapolation of single-label TTA methods (Han et al., 2025; Zhou et al., 2025; Zhang et al., 2025) proves inadequate, as these approaches fundamentally overlook the **label correlations** that are critical for multi-label tasks. On the other hand, specialized multi-label methods exemplified by BEM (Wu et al., 2025) are often impractical for real-time deployment, as they suffer from both high-latency iterative backpropagation and cumbersome preliminary preparations (e.g., external caption retrieval).

To address these limitations, we revisit multi-label test-time adaptation for frozen VLMs from a Bayesian perspective and deliberately avoid any modification of the pretrained backbone or prompts. Existing TTA approaches for VLMs typically adapt model parameters or prompts to the target distribution (Wu et al., 2024; Tan et al., 2025; Huang et al., 2025), which requires iterative backpropagation, increases

---

[*]Equal contribution  [1] State Key Laboratory of Novel Software Technology, Nanjing University, Nanjing 210023, China . Correspondence to: Zhiwei Jiang <jzw@nju.edu.cn>.

*Proceedings of the 43rd International Conference on Machine Learning*, Seoul, South Korea. PMLR 306, 2026. Copyright 2026 by the author(s).

latency and memory cost (Wang et al., 2025), and may overfit noisy test-time statistics, thereby eroding the strong zero-shot generalization of the original model. In contrast, we observe that the zero-shot scores of a VLM can be interpreted as marginal posterior signals produced by a fixed image–text alignment mechanism, while the degradation under distribution shift arises primarily from a mismatched prior over labels and their co-occurrence structure. This observation suggests a simple yet effective strategy: rather than tuning the model, we correct predictions by updating only the label prior, upgrading it from an unconditional prior to a conditional prior that explicitly captures label dependencies. Concretely, we propose Bayesian Conditional Priors (BCP), a backpropagation-free test-time adaptation framework for multi-label recognition with frozen vision–language models. Given a test image, BCP requires only a single forward pass of the pretrained VLM to obtain zero-shot logits, from which it selects a high-confidence label as an *Anchor*. Conditioning on this anchor, BCP applies a lightweight prior correction to the remaining logits to form an anchored posterior that amplifies semantically compatible labels and suppresses incompatible ones. This purely prior-based update preserves the strong zero-shot representation of the VLM, incurs negligible computational overhead, and is naturally suited to real-time multi-label TTA.

Importantly, this refinement admits a closed-form solution in logit space: each logit is adjusted by an additive term equal to the difference between a conditional and a marginal prior, which naturally corresponds to a pointwise mutual information (PMI) correction. Positive PMI increases the scores of labels that co-occur with the anchor more frequently than predicted by independence, while negative PMI down-weights labels that are statistically inconsistent with it. Despite its Bayesian grounding, BCP is extremely simple to implement. Conditional priors are estimated online from the unlabeled test stream using only running first- and second-order statistics of past model posteriors, maintained with constant memory and updated in closed form. In practice, the entire adaptation logic amounts to roughly ten lines of code, making BCP a plug-and-play module with negligible computational overhead.

Our main contributions can be summarized as:

- We identify label-independence as a fundamental limitation of lightweight multi-label TTA for frozen VLMs under shift, explaining characteristic suppression and incoherence failure modes.

- We introduce BCP, an anchor-conditioned Bayesian refinement that injects label dependencies into zero-shot predictions without tuning the backbone.

- We derive a closed-form logit correction that admits an information-theoretic PMI interpretation, and

we develop a practical online estimator of anchor-conditioned priors from unlabeled streams via second-order co-occurrence statistics, enabling stable streaming adaptation with negligible overhead.

Empirically, BCP yields substantial improvements on multi-label classification. With a ViT-B/16 backbone, it boosts mAP over CLIP by +7.1 to +11.1 points across datasets. Moreover, compared to strong multi-label TTA baselines such as BEM (Wu et al., 2025), BCP eliminates backpropagation and achieves higher accuracy with lower latency, reaching 61.69 mAP with 0.12s adapting time, whereas BEM attains 51.58 mAP with 0.24s.

## 2. Related Work

**Vision-Language Models.** Vision-language models (VLMs) align images and texts in a shared embedding space, enabling strong cross-modal transfer and zero-shot recognition (Laurençon et al., 2024; Radford et al., 2021; Li et al., 2023). Training on web-scale image–text pairs yields general representations that can be reused by prompting at inference time. As a result, VLMs have been widely adopted for text-guided image classification (Lafon et al., 2024; Zhou et al., 2022b;a), open-vocabulary recognition (Lan et al., 2024; Ding et al., 2023), and cross-modal retrieval (Wang et al., 2024; Guo et al., 2025). These properties make VLMs a natural backbone for deployment under distribution shift, where labeled target data is unavailable.

**Multi-Label Image Recognition.** Existing approaches for adapting VLMs to multi-label recognition generally fall into two distinct paradigms: fine-tuning and external knowledge-enhanced zero-shot adaptation. The first category, despite freezing the VLM backbone, necessitates labeled data to optimize auxiliary learnable parameters. For instance, DualCoOp (Sun et al., 2022) adapts the model by utilizing multi-label annotations to learn class-specific positive and negative prompt contexts. Similarly, T2I-PAL (Feng et al., 2025) bridges the modality gap by leveraging text-generated images, efficiently enhancing recognition through integrated prompting and adapter mechanisms. While effective, these methods rely on offline training with supervision, limiting their flexibility in data-scarce scenarios. The second category aims to boost zero-shot performance by incorporating rich semantic priors from external knowledge such as LLMs. Methods such as CoMC (Liu et al., 2024a) and SPARC (Miller et al., 2025) employ LLMs to generate comprehensive descriptions or structured templates, which are then used to guide the frozen VLM.

**Test-time adaptation.** Online test-time adaptation (TTA) for VLMs includes both gradient-based and lightweight approaches. TPT (Shu et al., 2022) adapts text prompts by entropy minimization, and DiffTPT (Feng et al., 2023)

further leverages diffusion-based augmentation for prompt tuning. While effective, these methods require backpropagation and iterative optimization, leading to noticeable latency and resource cost. To reduce overhead, TDA (Karmanov et al., 2024) performs fast adaptation with a cache of representative test samples, and Boostadapter (Zhang et al., 2024b) improves cache quality via region-guided augmentation. Another related line estimates test-time distributions: DOTA (Han et al., 2025) continuously updates distribution statistics, and ADAPT (Zhang et al., 2025) uses a sample bank to stabilize updates under shifts. BayesianTTA (Zhou et al., 2025) is also closely related in that it incorporates prior information for adaptation. Dedicated research on multi-label test-time adaptation (ML-TTA) scenarios remains scarce. To our knowledge, Bound Entropy Minimization (BEM) (Wu et al., 2025) is the only existing ML-TTA method aligned with our core objective: improving the reliability of multiple high-confidence predictions per sample. However, its implementation is complex—it requires prefetching paired textual captions for each test image to determine the label count and involves iterative back-propagation updates to the prompt parameters (Zanella & Ben Ayed, 2024).

In contrast, our approach is strictly gradient-free and requires no preliminary preparation.

## 3. Method

### 3.1. Preliminaries: Zero-shot CLIP Posterior

We build on a pre-trained CLIP model that is trained to align visual and textual representations on large-scale image–text pairs via contrastive learning. CLIP consists of a visual encoder $\boldsymbol{f}^v(\cdot)$ and a text encoder $\boldsymbol{f}^t(\cdot)$ that map images and class prompts into a shared feature space. Let $x$ be an input image $\mathcal{C} = \{c_1, \ldots, c_K\}$ denote the set of $K$ target classes, where each $c_k \in \{0, 1\}$ is a binary variable indicating the presence of a class. For zero-shot prediction, each class $c_k$ is associated with a textual prompt $t_k$ (e.g., "a photo of a {class}"). Given an image $x$, we first compute the zero-shot logits vector $\mathbf{s} \in \mathbb{R}^K$ via a scaled cosine similarity. For class $c_k$, its logit value $s_k$ is defined as:

$$s_k = \frac{\langle \boldsymbol{f}^v(x), \boldsymbol{f}^t(t_k) \rangle}{\tau}, \qquad (1)$$

where $\tau$ is the learned temperature and both embeddings are $\ell_2$-normalized. Then we use the Softmax function to yield the zero-shot posterior over classes:

$$P_{\mathrm{zs}}(c_k = 1 \mid x) = \frac{\exp(s_k)}{\sum_{j=1}^{K} \exp(s_j)}. \qquad (2)$$

We interpret $P_{\mathrm{zs}}(c_k = 1 \mid x)$ as CLIP's zero-shot posterior for $c_k$,, which serves as an initial estimate to be refined

by our Bayesian conditional prior estimation. For each test sample $x$, we denote its CLIP zero-shot posterior by $\mathbf{p} = \big[ P_{\mathrm{zs}}(c_1 = 1 \mid x), \ldots, P_{\mathrm{zs}}(c_K = 1 \mid x) \big]$.

### 3.2. Multi-label Bayesian Posterior with Conditional Priors

To overcome the mutual exclusivity bias inherent in the zero-shot posterior, we propose a Bayesian framework that explicitly models the semantic correlations among classes. Firstly, we derive the posterior for class $c_k$ via Bayes' theorem:

$$P(c_k = 1 \mid x) = \frac{P(x \mid c_k = 1)\, P(c_k = 1)}{P(x)}, \qquad (3)$$

where $P(x \mid c_k = 1)$ is the visual likelihood conditioned on $c_k$ and $P(c_k = 1)$ is the marginal prior. This equation interprets the CLIP prediction as the product of visual evidence and prior belief. Conceptually, the term $P(x \mid c_k = 1)$ captures the semantic alignment between the image and the class text, while $P(c_k = 1)$ represents the baseline probability of the class occurring.

To explicitly model label correlations, we condition the inference on a high-confidence *anchor* class $c_a$. We formulate the conditional posterior as: $P(c_k = 1 \mid x, c_a = 1) \propto P(x \mid c_k = 1, c_a = 1)\, P(c_k = 1 \mid c_a = 1)$. Here, a critical challenge lies in the joint likelihood term. We posit that the visual manifestation of class $c_k$ depends primarily on its intrinsic features rather than the presence of the anchor $c_a$. Based on this intuition, we introduce a conditional independence assumption regarding visual likelihood: $P(x \mid c_k = 1, c_a = 1) \approx P(x \mid c_k = 1)$. This implies that label dependencies are captured solely by the conditional prior $P(c_k \mid c_a)$, while the visual evidence remains disentangled. Consequently, the conditional posterior simplifies to $P(c_k = 1 \mid x, c_a = 1) \propto P(x \mid c_k = 1)\, P(c_k = 1 \mid c_a = 1)$. By substituting the likelihood term $P(x \mid c_k = 1)$ derived from Eq. (3), we arrive at the final anchored posterior:

$$P(c_k = 1 \mid x, c_a = 1) \propto \underbrace{P(c_k = 1 \mid x)}_{\text{Original Posterior}}$$
$$\times \underbrace{\left( \frac{P(c_k = 1 \mid c_a = 1)}{P(c_k = 1)} \right)}_{\text{Bayesian Calibration Factor}}. \qquad (4)$$

This factorization reveals a rigorous adjustment mechanism: the zero-shot prediction is modulated by a Bayesian calibration factor, defined as the ratio between the conditional prior and the marginal prior. This ratio effectively quantifies the semantic affinity between the target class and the anchor.

To implement the correction efficiently, we operate directly in logit space. Interpreting the zero-shot CLIP logit $s_k$

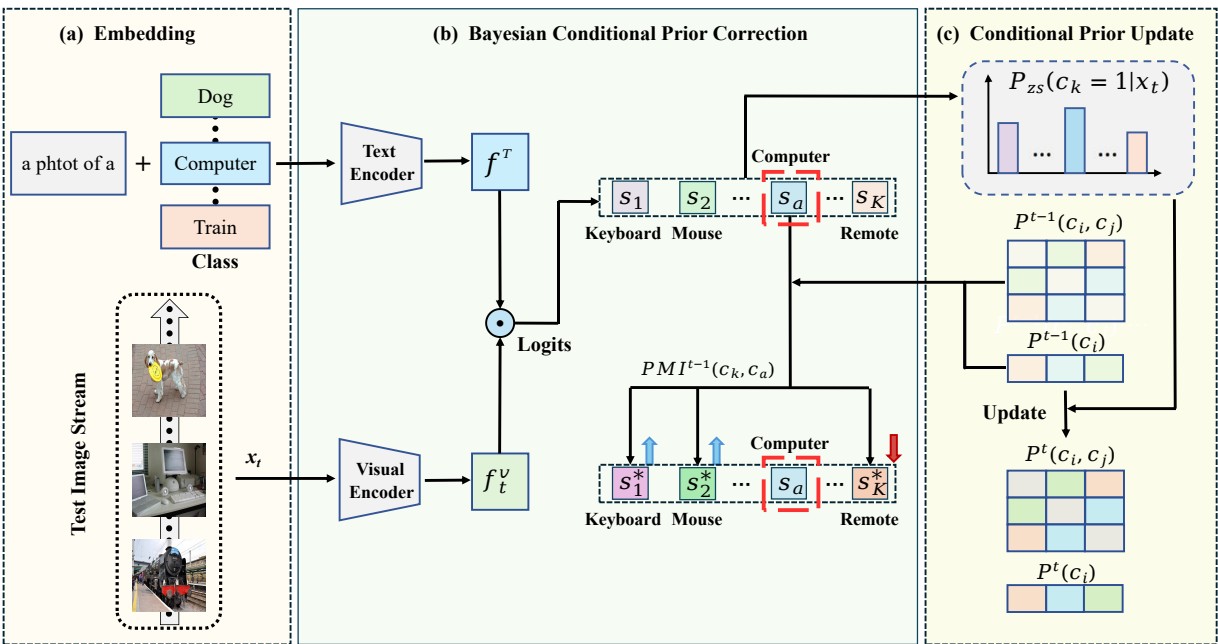

*Figure 2.* Overview of our Bayesian Conditional Priors (BCP) Estimation for zero-shot multi-label CLIP. For each test image $x_t$, frozen CLIP encoders score class prompts to produce zero-shot logits and the posterior $P(c \mid x_t)$. We select an anchor label $a$ with the highest confidence and apply a closed-form logit correction using online conditional priors $P(c_i = 1 \mid c_a = 1)$, yielding $P(c \mid x_t, c_a = 1)$ and injecting pairwise label dependencies without updating CLIP. The conditional priors are updated online from past predictions via running marginals $P^t(c_i)$ and co-occurrences $P^t(c_i, c_j)$, enabling continual adaptation to the target test distribution with negligible overhead.

as an unnormalized log-posterior, it can be conceptually decomposed as:

$$s_k = \log P(x \mid c_k = 1) + \log P(c_k = 1) + \mathrm{C}, \quad (5)$$

where C is a class-independent constant. This formulation allows us to view the original logits as implicitly containing a marginal class prior. Next, taking the logarithm of the anchored posterior in Eq. (4), we derive the additive update rule for the corrected logit $s_k^*$:

$$
\begin{aligned}
s_k^* &= \log P(c_k = 1 \mid x, c_a = 1) + \mathrm{C} \\
&= s_k + \underbrace{\left( \log P(c_k = 1 \mid c_a = 1) - \log P(c_k = 1) \right)}_{\Delta_{\text{prior}}}.
\end{aligned}
$$
$$(6)$$

Here, $\Delta_{\text{prior}}$ represents the log-space correction induced by label dependency. Ideally, this term corresponds to the Pointwise Mutual Information (PMI) (Bouma, 2009) between the target class $k$ and the anchor $a$, effectively biasing the decision boundary based on semantic co-occurrence.

### 3.3. Bayes-guided Test-time Adaptation

Translating the theoretical framework from Sec. 3.2 into a practical TTA protocol requires addressing two challenges:

(1) approximating statistics (i.e., the marginal priors $P(c_k)$ and conditional priors $P(c_k = 1 \mid c_a = 1)$) from the incoming test stream, and (2) achieving effective adaptation with negligible computational overhead by bypassing the iterative gradient updates typically required in TTA.

To approximate population-level statistics from the streaming test data, we maintain running estimates of the second-order co-occurrence. Let $\mathbf{p}^{(t)} \in \mathbb{R}^K$ denote the zero-shot posterior for the current sample $x_t$. We define the entries of the global co-occurrence matrix $\mathbf{U}^t$ and marginal vector $\mathbf{m}^t$ accumulated up to step $t$ as:

$$U_{j,k}^t = \sum_{i=1}^{t} p_j^{(i)} p_k^{(i)}, \quad m_k^t = \sum_{i=1}^{t} p_k^{(i)}, \quad (7)$$

where $p_k^{(i)} = P_{\text{zs}}(c_k = 1 \mid x)$. During inference for $x_t$, we strictly adhere to a causal protocol by utilizing only the *historical* statistics from the previous step $t-1$. For $x_t$, we identify its anchor class $a = \arg\max_k p_k^{(t)}$. If the anchor confidence is reliable ($p_a^{(t)} > \mu$), we compute the conditional prior and marginal probability using the historical states:

$$P^t(c_k = 1 \mid c_a = 1) \approx \frac{U_{k,a}^{t-1}}{m_a^{t-1} + \varepsilon}, \quad (8)$$
$$P^t(c_k = 1) \approx m_k^{t-1}/(t-1)$$

---

**Algorithm 1** Bayesian Conditional Priors Estimation

---

1: **Input:** Frozen CLIP model $\mathcal{M}$, test stream $\{x_t\}_{t=1}^N$, threshold $\mu$.
2: **Initialize:** $\mathbf{U} \leftarrow \mathbf{0}_{K \times K}$, $\mathbf{m} \leftarrow \mathbf{0}_K$.
3: **for** $t = 1$ **to** $N$ **do**
4:    // 1. Zero-shot Inference
5:    Compute logits $\mathbf{s}^{(t)}$ and probs $\mathbf{p}^{(t)}$ via $\mathcal{M}(x_t)$.
6:    Identify anchor $a \leftarrow \arg\max_k \mathbf{p}^{(t)}$.
7:    Initialize corrected logits $\mathbf{s}^{*(t)} \leftarrow \mathbf{s}^{(t)}$.
8:    // 2. Bayesian Correction
9:    **if** $p_{\text{zs},a}^{(t)} > \mu$ **and** $t > 1$ **then**
10:       Estimate priors using $\mathbf{U}, \mathbf{m}$ via Eq. (8).
11:       Compute correction term $\Delta_{\text{prior}}$.
12:       Update logits $\mathbf{s}^{*(t)} \leftarrow \mathbf{s}^{(t)} + \Delta_{\text{prior}}$ via Eq. (9).
13:    **end if**
14:    **Output:** Final prediction based on $\mathbf{s}^{*(t)}$.
15:    // 3. Online Statistics Update
16:    Update $\mathbf{U} \leftarrow \mathbf{U} + \mathbf{p}^{(t)}(\mathbf{p}^{(t)})^\top$.
17:    Update $\mathbf{m} \leftarrow \mathbf{m} + \mathbf{p}^{(t)}$.
18: **end for**

---

where $\varepsilon$ is a stability constant added to the denominator to prevent division by zero.

Finally, we obtain the Bayes-corrected logits $\mathbf{s}^{(t)}$ at step $t$, the logit $s_k^{*(t)}$ for $c_k$ is updated as:

$$
s_k^{*(t)} = \begin{cases} s_k^{(t)}, & t = 1, \\ s_k^{(t)} + \log \dfrac{U_{k,a}^{t-1}}{m_a^{t-1} + \varepsilon} - \log \dfrac{m_k^{t-1}}{t-1}, & t > 1. \end{cases}
$$
(9)

Note that $k \neq a$: once the anchor is fixed, we do not apply any correction to it. Ideally, the correction applies only for $t > 1$; for the first sample, the model operates in a pure zero-shot manner.

Finally, we update the global statistics using the zero-shot posterior $\mathbf{p}^{(t)}$:

$$
\mathbf{U}^t \leftarrow \mathbf{U}^{t-1} + \mathbf{p}^{(t)}(\mathbf{p}^{(t)})^\top, \quad \mathbf{m}^t \leftarrow \mathbf{m}^{t-1} + \mathbf{p}^{(t)}. \quad (10)
$$

These closed-form updates are **gradient-free**, ensuring negligible computational overhead. The complete procedure is outlined in Algorithm 1.

Importantly, BCP does not create a cross-sample self-reinforcing error accumulation loop through the PMI correction itself. The running marginal and co-occurrence statistics are updated only from the original zero-shot CLIP posterior $\mathbf{p}^{(t)}$, rather than from the PMI-corrected logits $\mathbf{s}^{*(t)}$ or the final predictions. Therefore, an incorrect correction on one sample is not fed back into the online statistics and does not recursively amplify itself over future samples. The more relevant issue is instead intra-sample miscorrection when the selected anchor is unreliable; BCP reduces

this risk by applying the correction only when the anchor confidence exceeds $\mu$, and we empirically examine anchor misses in Sec. 4.2.

### 3.4. Proof: PMI View of the Correction.

Observe that the prior correction term in Eq. (6) satisfies

$$
\begin{aligned}
\Delta_{\text{prior}} &= \log P(c_k = 1 \mid c_a = 1) - \log P(c_k = 1) \\
&= \log \frac{P(c_k = 1, c_a = 1)}{P(c_k = 1)\,P(c_a = 1)},
\end{aligned}
$$
(11)

which is exactly the Pointwise Mutual Information (PMI) between events $\{c_k = 1\}$ and $\{c_a = 1\}$. Therefore, our update can be written as

$$
s_k^* = s_k + \text{PMI}(c_k, c_a). \quad (12)
$$

This PMI term admits two equivalent interpretations. *(i) Log Bayes factor:*

$$
\Delta_{\text{prior}} = \log \frac{P(c_k = 1 \mid c_a = 1)}{P(c_k = 1)}, \quad (13)
$$

which compares the plausibility of label $k$ under the conditional prior induced by the anchor versus the marginal prior; positive/negative values respectively promote/suppress $k$. *(ii) Surprisal reduction:* $-\log P(c_k = 1)$ is the surprisal of label $k$, while $-\log P(c_k = 1 \mid c_a = 1)$ is the conditional surprisal after observing the anchor; their difference (PMI) is the specific information that $c_a$ provides about $c_k$.

Moreover, PMI is consistent with a standard information-gain view in expectation. Let $\Delta_{\text{prior}}(c_k) \triangleq \log P(c_k \mid c_a = 1) - \log P(c_k)$, whose instance at $c_k = 1$ recovers our update term. Then

$$
\begin{aligned}
&\mathbb{E}_{c_k \sim P(\cdot \mid c_a = 1)}\big[\Delta_{\text{prior}}(c_k)\big] \\
&= D_{\text{KL}}\big(P(c_k \mid c_a = 1) \,\|\, P(c_k)\big) \geq 0,
\end{aligned}
$$
(14)

i.e., conditioning on a reliable anchor shifts the prior belief about $c_k$ away from its marginal by a nonnegative amount of information. Averaging further over $c_a$ yields the mutual information:

$$
\begin{aligned}
I(c_k; c_a) &= \mathbb{E}_{c_a}\, D_{\text{KL}}\big(P(c_k \mid c_a) \,\|\, P(c_k)\big) \\
&= \mathbb{E}_{c_k, c_a}\big[\text{PMI}(c_k, c_a)\big].
\end{aligned}
$$
(15)

Taken together, Eq. (6) injects a pairwise dependency potential (PMI) into the logit while keeping the frozen CLIP-induced likelihood unchanged, yielding a closed-loop interpretation: the Bayesian ratio in Eq. (4) is an additive information term that transfers label-dependency knowledge from the target stream into prediction without modifying the encoders.

When several non-anchor labels have nearly identical co-occurrence statistics with the anchor, their PMI corrections

become similar as well. In this case, their relative ranking is mainly determined by the original zero-shot visual evidence. This is a deliberate design choice: BCP calibrates the prior while preserving CLIP's frozen visual discrimination, rather than forcing an additional ranking from weak or nearly indistinguishable second-order statistics. Thus, the correction changes the ranking when the target-stream prior provides a clear compatibility signal, but leaves fine-grained discrimination to the pretrained CLIP logits when the co-occurrence evidence is similar.

### 3.5. Discussion: Why Softmax and a Single Anchor?

A natural query arises regarding the use of Softmax, as it introduces an inductive bias of mutual exclusivity that appears intuitively ill-posed for multi-label scenarios. We deliberately retain this formulation to preserve strict alignment with CLIP's pre-training InfoNCE objective, establishing it as the logical premise of our theoretical model. Specifically, while Softmax inherently enforces inter-class competition, this characteristic serves as the precise motivation for our framework: we aim to calibrate this competitive distribution by injecting correlation priors, thereby recovering label co-occurrence without disrupting the pre-trained feature alignment.

This choice also supports our adaptation mechanism. We select a single anchor $a = \arg\max_k p_k$ and update the prior only when $p_a > \mu$, which requires a posterior that is comparable across classes and yields a decisive top prediction. Independent sigmoid outputs often saturate under temperature scaled similarity logits and compress inter-class gaps, making confidence gating unreliable and weakening the notion of a strong anchor. Softmax assigns a fixed probability mass and sharpens relative evidence, so $p_a$ is a more meaningful reliability signal. We use only one anchor because our correction is conditioned on the event that the anchor is true and is modeled through pairwise terms $P(c_k \mid c_a)$. Multiple anchors would demand higher order priors or mix inconsistent conditions, increasing variance and noise sensitivity. Together, Softmax and a single high confidence anchor make the calibration selective, stable, and consistent with the pretrained representation. Ablations on multiple anchors are in Appendix A.3.

## 4. Experiment

### 4.1. Experimental Setup

**Benchmarks.** We adopt the widely used CLIP model (Radford et al., 2021) as the source model and evaluate on three standard multi-label benchmarks—VOC (Vicente et al., 2014), MSCOCO (Lin et al., 2014), and NUSWIDE (Chua et al., 2009)—as target domains. The VOC dataset comprises 20 categories and includes both VOC2007 and VOC2012, with 4,952 and 5,823 test images, respectively. MSCOCO extends the label space to 80 categories; following common practice, we use the COCO2014 validation set with 40,504 images and the COCO2017 validation set with 5,000 images for evaluation, since the official test annotations are not publicly available. The NUSWIDE dataset contains 81 categories and 83,898 test images of relatively low resolution, providing an even broader label spectrum than MSCOCO.

**Implementation Details.** All experiments are conducted with CLIP (Radford et al., 2021) as the backbone, using four standard variants: RN50, RN101, ViT-B/32, and ViT-B/16, each consisting of an image encoder paired with a text encoder. We use the CLIP prompt template "a photo of a" to construct class text prompts. Unless otherwise specified, the confidence threshold $\mu$ is fixed to $0.5$ for all benchmarks. In all settings, multi-label test-time adaptation is performed in a strictly online fashion, *i.e.*, the batch size is set to 1. We report mean Average Precision (mAP) as the primary evaluation metric, defined as $\text{mAP} = \frac{1}{L} \sum_{i=1}^{L} \text{AP}_i$, where $L$ denotes the number of categories and $\text{AP}_i$ is the area under the precision recall curve for the $i$-th category.

### 4.2. Comparisons with SOTA and Analysis.

We compare BCP with representative CLIP-based test-time adaptation baselines under both multi-label and single-label protocol in Sec. 4.1. Some results are consistent with BEM (Wu et al., 2025). Unless stated otherwise, we follow the recommended test-time budgets and default hyperparameters of each method, and report mAP on multiple target datasets and backbones.

**Results across architectures.** Table 1 shows that BCP delivers the best overall performance on both RN50 and ViT-B/16 and remains consistently strong across all five datasets. Compared with the SOTA method BEM (Wu et al., 2025), BCP achieves higher mAP on every dataset for both architectures, indicating that the proposed correction is consistently effective across backbones and label spaces. Among these baselines, SCA is the strongest competitor, reaching an average mAP of 69.24, while BCP further improves the average to 71.79 and obtains the best result on all five benchmarks. In contrast, several baselines adapted from single-label TTA provide only limited gains and may even degrade performance, suggesting that directly reusing single-label objectives is unreliable in the multi-label setting. DOTA performs reasonably well on COCO and VOC, where a nontrivial fraction of samples contain few active labels and better match its single-label oriented adaptation assumption. However, DOTA fails on NUSWIDE, where lower image quality and weaker initial CLIP predictions make the visual mean and class covariance estimates more susceptible to distribution shift, leading to unstable correc-

*Table 1.* Comparison with CLIP and SOTAs on adapting multi-label instances with different architectures.

| | METHOD | BP-FREE | COCO2014 | COCO2017 | VOC2007 | VOC2012 | NUSWIDE | AVERAGE |
|---|---|---|---|---|---|---|---|---|
| **RN-50** | CLIP (RADFORD ET AL., 2021) | √ | 47.53 | 47.32 | 75.91 | 74.25 | 41.53 | 57.31 |
| | DMN (ZHANG ET AL., 2024C) | √ | 44.54 | 44.18 | 74.87 | 74.13 | 41.32 | 55.81 |
| | TDA (KARMANOV ET AL., 2024) | √ | 48.91 | 49.11 | 76.64 | 75.12 | 42.34 | 58.42 |
| | TPT (SHU ET AL., 2022) | × | 48.52 | 48.51 | 75.54 | 73.92 | 41.97 | 57.69 |
| | DiffTPT (FENG ET AL., 2023) | × | 48.56 | 48.67 | 75.89 | 74.13 | 41.33 | 57.72 |
| | RLCF (ZHAO ET AL., 2024) | × | 36.87 | 36.73 | 65.75 | 64.73 | 29.83 | 46.78 |
| | DOTA (HAN ET AL., 2025) | √ | 55.69 | 59.58 | 84.55 | 82.17 | 38.88 | 64.17 |
| | BEM (WU ET AL., 2025) | × | 51.58 | 51.39 | 78.62 | 76.63 | 42.53 | 60.15 |
| | **BCP** | √ | **61.69** | **61.65** | **86.93** | **85.51** | **50.34** | **69.22** |
| **ViT-B/16** | CLIP (RADFORD ET AL., 2021) | √ | 54.42 | 54.13 | 79.58 | 79.25 | 45.65 | 62.61 |
| | DMN (ZHANG ET AL., 2024C) | √ | 52.52 | 52.37 | 79.83 | 79.67 | 46.27 | 62.13 |
| | DART (LIU ET AL., 2024B) | × | 54.73 | 54.68 | 79.91 | 78.56 | 45.91 | 62.76 |
| | TDA (KARMANOV ET AL., 2024) | √ | 55.21 | 55.46 | 80.12 | 79.92 | 46.72 | 63.49 |
| | TPT (SHU ET AL., 2022) | × | 53.32 | 54.20 | 77.54 | 77.39 | 46.15 | 61.72 |
| | DiffTPT (FENG ET AL., 2023) | × | 53.91 | 54.15 | 77.93 | 77.24 | 46.13 | 61.87 |
| | RLCF (ZHAO ET AL., 2024) | × | 54.21 | 54.43 | 79.29 | 79.26 | 43.18 | 62.07 |
| | DOTA (HAN ET AL., 2025) | √ | 56.29 | 62.89 | 84.97 | 82.86 | 32.74 | 63.95 |
| | BEM (WU ET AL., 2025) | × | 57.52 | 57.49 | 81.28 | 81.13 | 46.55 | 64.80 |
| | STS (DAFNIS & METAXAS, 2026) | × | 59.80 | 59.09 | 85.09 | 83.73 | 47.23 | 66.99 |
| | ADAPT (ZHANG ET AL., 2025) | √ | 61.91 | 59.84 | 86.34 | 84.92 | 43.18 | 67.24 |
| | SSG (ZHOU ET AL., 2026) | √ | 61.49 | 60.96 | 86.02 | 84.53 | 47.93 | 68.19 |
| | SCA (GUAN ET AL., 2026) | √ | 63.31 | 61.81 | 86.99 | 86.00 | 48.08 | 69.24 |
| | **BCP** | √ | **65.47** | **64.92** | **88.41** | **87.37** | **52.78** | **71.79** |

*Table 2.* Impact of anchor correctness on mAP across datasets. Results are reported on the subset of ($P_{\max} > \mu$). "Hit" denotes that the predicted anchor is in the ground-truth labels, while "Miss" denotes otherwise. The last two rows summarize the composition of the subset.

| | SETTING | METHOD | COCO2014 | COCO2017 | VOC2007 | VOC2012 | NUSWIDE | AVERAGE |
|---|---|---|---|---|---|---|---|---|
| **ViT-B/16** | ANCHOR-HIT | CLIP | 61.64 | 62.28 | 85.10 | 84.24 | 70.38 | 72.73 |
| | ANCHOR-HIT | BCP | **69.14** | **68.90** | **92.18** | **90.49** | **80.96** | **80.33** |
| | ANCHOR-MISS | CLIP | 22.35 | 26.91 | **24.94** | **23.91** | 12.65 | **22.15** |
| | ANCHOR-MISS | BCP | **23.83** | **27.98** | 21.19 | 20.41 | **13.20** | 21.32 |
| | HIT SHARE (%) | - | 93.88 | 93.69 | 89.15 | 91.63 | 58.53 | 85.38 |
| | MISS SHARE (%) | - | 6.12 | 6.31 | 10.85 | 8.37 | 41.47 | 14.62 |

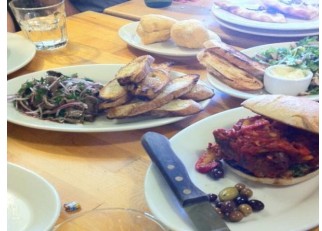
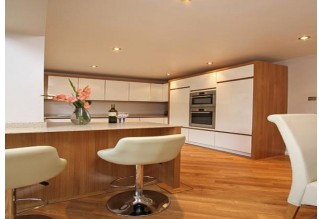
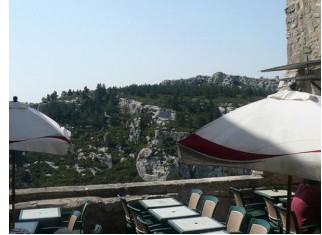
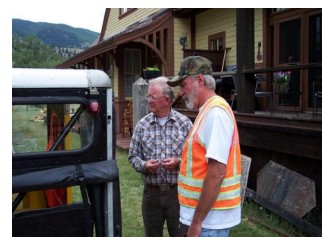

*(a)* Anchor (Knife): Fork.  *(b)* Anchor (Oven): Refrigerator.  *(c)* Anchor (Table): Aeroplane.  *(d)* Anchor (Chair): Bus.

*Figure 3.* Case Study of Anchor-Miss Samples from PASCALVOC and MSCOCO.

tion signals and reduced accuracy. Overall, these results suggest that explicitly correcting the label prior with on-line anchor-conditioned co-occurrence statistics provides complementary information that existing cache-, bank-, or optimization-based TTA methods do not fully exploit in

multi-label recognition. More results across architectures are provided in appendix A.1.

**Robustness to test-stream ordering.** Because BCP esti-mates conditional priors online, we further evaluate whether its performance depends on a particular test-sample order-

*Table 3.* Results under five random orderings of the test stream on ViT-B/16.

| DATASET | SEED1 | SEED2 | SEED3 | SEED4 | SEED5 | AVG. |
|---------|-------|-------|-------|-------|-------|------|
| COCO14  | 65.48 | 65.40 | 65.48 | 65.49 | 65.42 | 65.45 |
| COCO17  | 64.85 | 64.79 | 64.93 | 64.85 | 64.76 | 64.84 |
| VOC07   | 88.31 | 88.21 | 88.31 | 88.19 | 88.24 | 88.25 |
| VOC12   | 87.38 | 87.42 | 87.41 | 87.29 | 87.36 | 87.37 |
| NUS     | 52.65 | 52.71 | 52.89 | 52.92 | 52.74 | 52.78 |

*Table 4.* Results on different label count.

| METHOD | $\{1, 2\}$ | $\{3, 4\}$ | $\{5, 6, 7\}$ | $\{\geq 8\}$ |
|--------|------------|------------|---------------|--------------|
| CLIP    | 62.76 | 55.41 | 49.89 | 41.07 |
| TPT     | 62.88 | 53.05 | 45.57 | 37.43 |
| DIFFTPT | 61.97 | 52.67 | 44.32 | 36.89 |
| RLCF    | 66.01 | 51.65 | 43.32 | 35.08 |
| BEM     | 67.14 | 57.59 | 51.68 | 41.32 |
| **BCP** | **73.85** | **66.26** | **61.72** | **51.64** |

ing. We repeat evaluation with five random permutations of each test stream using the ViT-B/16 backbone. As shown in Table 3, the results are highly consistent across different seeds on all benchmarks, indicating that BCP is robust to random test-stream ordering under the standard shuffled online protocol.

**Analysis of the impact of anchor.** We analyze anchor reliability in BCP on the confident subset $P_{\max} > \mu$ by splitting predictions into *Anchor-Hit* and *Anchor-Miss*, depending on whether the predicted anchor is present in the ground-truth labels. This decomposition separates the impact of the prior correction that is conditioned on the anchor when the assumption $c_a = 1$ holds. It also makes clear how the correction behaves when this assumption does not hold. Table 2 shows that under *Anchor-Hit*, BCP consistently beats frozen CLIP on every benchmark. The mean mAP rises from 72.73 to 80.33. The improvement is 7.50 mAP on COCO2014 and 10.58 mAP on NUSWIDE. These results support that the co-occurrence prior conditioned on the anchor is effective when $c_a = 1$. Under *Anchor-Miss*, as Table 2 shows that BCP improves CLIP by 1.48 mAP on COCO2014, 1.07 mAP on COCO2017, and 0.55 mAP on NUSWIDE, but it drops by 3.75 mAP on VOC2007 and 3.50 mAP on VOC2012. We analyze this phenomenon as follows. On COCO and VOC, the confident subset is dominated by *Anchor-Hit*, the Hit share is above 89%, so the overall improvement is mainly driven by samples with correct anchors. NUSWIDE shows a different pattern. The Hit share is 58.53%, yet BCP greatly improves *Anchor-Hit* from 70.38 to 80.96 mAP, and it also slightly improves *Anchor-Miss* from 12.65 to 13.20 mAP. And BCP still delivers a 7.18 mAP gain on the full dataset. This means the overall gain is mainly driven by stronger *Anchor-Hit* performance, while the many *Anchor-Miss* samples do not offset it. This is consistent with our case study: with large label sets, many misses are still semantically close to the ground-truth labels, so the prior conditioned on the anchor remains useful. Additional ablations are in Appendix A.2.

**Analysis on head, medium, and tail label splits.** To examine whether the proposed conditional-prior correction suppresses low-frequency categories, we split the labels of each dataset into head, medium, and tail groups according to class frequency, with categories evenly divided from the

*Table 5.* Performance under head, medium, and tail label splits.

| DATASET | METHOD | HEAD | MEDIUM | TAIL |
|---------|--------|------|--------|------|
| COCO14 | CLIP | 42.47 | 62.53 | 57.93 |
|        | SCA  | 48.03 | 74.08 | 67.61 |
|        | **BCP** | **51.84** | **75.51** | **69.22** |
| COCO17 | CLIP | 42.29 | 60.45 | 59.51 |
|        | SCA  | 47.17 | 70.10 | 68.42 |
|        | **BCP** | **51.42** | **73.08** | **70.29** |
| VOC07  | CLIP | 80.15 | 70.61 | 89.30 |
|        | SCA  | 86.33 | 81.53 | 94.12 |
|        | **BCP** | **88.13** | **82.56** | **95.57** |
| VOC12  | CLIP | 80.42 | 71.78 | 87.50 |
|        | SCA  | 86.04 | 80.51 | 92.38 |
|        | **BCP** | **87.58** | **81.60** | **93.80** |
| NUS    | CLIP | 47.57 | 43.15 | 46.01 |
|        | SCA  | 48.20 | 47.66 | 48.37 |
|        | **BCP** | **52.47** | **47.87** | **55.90** |

most frequent to the least frequent. Table 5 reports the mean AP within each group. BCP consistently outperforms both CLIP and SCA across all three groups on all five benchmarks, including the tail split. This result does not support the concern that BCP systematically suppresses rare labels; instead, it indicates that anchor-conditioned prior calibration remains effective for low-frequency classes, where useful co-occurrence evidence can help recover labels that are weakly expressed by the independent zero-shot posterior.

**Case Study of Anchor-Miss Samples.** Fig. 3 suggests that *Anchor-Miss* is not a single failure mode, but splits into two qualitatively different cases. The anchor outside the parentheses is incorrect; the label inside the parentheses is the ground-truth class it is confused with. On large labelset datasets like COCO2014, COCO2017 and NUS-WIDE, even when the predicted anchor label is absent from the ground-truth set, it often correctly captures the high-level semantic scene. For example, the model might mistake a oven for a refrigerator, or predict a different kind of tableware in a dining scene. These predictions, though incorrect in specific class label, still convey the correct high-level semantic information (e.g., both "oven" and "refrigerator" indicate the presence of a kitchen). As a result, the conditional co-occurrence prior built from such anchors remains highly compatible with the other true labels at the semantic

*Table 6.* Comparison With More Zero-shot Multi-label Classification Methods.

| METHOD | FT-FREE | EKE-FREE | oTTA | VOC07 | NUS |
|--------|---------|----------|------|-------|------|
| CoMC | × | × | × | 89.40 | 48.20 |
| **BCP** | √ | √ | √ | 86.93 | 50.34 |
| SPARC | √ | × | × | 88.70 | 47.30 |
| **BCP** | √ | √ | √ | 88.41 | 52.78 |

*Table 7.* Results on adaptation complexity.

| METHOD | TPT | DIFFTPT | RLCF | BEM | **BCP** |
|--------|-----|---------|------|-----|---------|
| BP-FREE | × | × | × | × | √ |
| TIME ↓ | 0.21s | 0.41s | 0.45s | 0.24s | **0.112s** |
| MAP ↑ | 48.52 | 48.56 | 36.87 | 51.58 | **61.69** |

level.

While VOC2007/VOC2012 behaves differently. Because its label set is smaller and more distinct, a wrong anchor is more likely to come from a different semantic scene (e.g., mixing road vehicles with aircraft). In this setting, the prior conditioned on the anchor may no longer align well with the image. It can put extra weight on co-occurrences that are not supported by the visual content and reduce the scores of correct labels, which can hurt performance. Overall, the case study indicates that the impact of *Anchor-Miss* is directly determined by the semantic distance between the predicted anchor and the true label set.

**Comparison With More Zero-shot Multi-label Classification Methods.** We further compare BCP with representative zero-shot multi-label recognition methods based on frozen VLMs, including CoMC (Liu et al., 2024a) and SPARC (Miller et al., 2025). These methods typically require additional module fine-tuning (FT) or external knowledge enhancement (EKE). For example, SPARC relies on LLMs to generate a series of combined class prompts. Additionally, these methods do not support strict online (oTTA) deployment with a batch size of 1. Table 6 highlights that BCP achieves competitive accuracy while meeting stricter deployment constraints. With RN50, BCP is slightly below CoMC on VOC2007, yet it surpasses CoMC on NUSWIDE from 48.20 to 50.34. With ViT-B/16, BCP closely matches SPARC on VOC2007, reaching 88.41 versus 88.70, and it exceeds SPARC by a clear margin on NUS, improving from 47.30 to 52.78. Overall, the results show that BCP preserves strong performance and delivers the best NUS accuracy in both settings, while remaining fine-tuning free, external knowledge free, and compatible with strict online test-time adaptation at batch size one.

**Results with Different Label Counts on COCO2014.** Table 7 compares adaptation complexity and effectiveness across five test time adaptation methods. BCP avoids backpropagation, which reduces optimization overhead and improves deployment practicality. BCP also achieves the lowest per sample adaptation time at 0.112s, which is nearly two times faster than TPT and more than three times faster than DiffTPT and RLCF. In addition to efficiency, BCP delivers the best detection accuracy with 61.69 mAP, outperforming the strongest baseline BEM by 10.11 mAP and

exceeding TPT and DiffTPT by more than 13 mAP. These results indicate that BCP provides a favorable tradeoff between adaptation cost and performance, enabling fast and reliable test time adaptation.

**Results on adaptation complexity.** Table 7 evaluates the accuracy latency trade-off of test-time adaptation on COCO2014 with CLIP-ResNet50. Iterative methods are substantially slower: BEM attains 51.58 mAP at 0.24 s, while DiffTPT/RLCF are heavier without improving accuracy. In contrast, BCP achieves the highest accuracy with low overhead, about $2\times$ faster than BEM and faster than TPT with a +13.17 mAP gain. This efficiency stems from avoiding test-time gradient updates: BCP applies lightweight statistics, so runtime is dominated by a single forward pass rather than backpropagation loops, making it suitable for latency-critical deployment.

## 5. Conclusion

We studied test-time adaptation for zero-shot CLIP in realistic multi-label settings, where label dependencies matter but are largely ignored by existing statistics-based methods. From a Bayesian view of CLIP zero-shot scores, we proposed Bayesian Conditional Priors (BCP) Estimation, a gradient-free refinement framework that injects label dependencies through anchor-conditioned priors estimated online from second-order co-occurrence statistics. BCP runs in the forward pass, requires no backpropagation or external retrieval, and maintains only lightweight label-space state. Future work includes extending priors beyond pairwise structure, improving online estimation under concept drift and miscalibrated posteriors, and generalizing to open-vocabulary multi-label detection and segmentation.

## Acknowledgements

We would like to thank the anonymous reviewers for their insightful comments. This work is supported by the JiangSu Natural Science Foundation under Grant No.BK20251989; the National Natural Science Foundation of China under Grants Nos. 62172208, 62441225, 61972192; the Fundamental and Interdisciplinary Disciplines Breakthrough Plan of the Ministry of Education of China (No.JYB2025XDXM118); the "111 Center" (No. B26023). This work is partially supported by Collabora-

tive Innovation Center of Novel Software Technology and Industrialization.

## Impact Statement

This work aims to advance research in machine learning, with a focus on improving the robustness and reliability of vision-language models in real-world open environments. We propose an efficient, backpropagation-free test-time adaptation method (BCP) that calibrates multi-label predictions by explicitly modeling label dependencies during inference. Our approach does not modify the parameters of the pre-trained model and relies only on online statistics from the test stream, resulting in minimal computational overhead and ease of deployment. From an ethical and social perspective, this research does not involve sensitive data nor does it introduce new ethical concerns. The primary contribution lies in methodological lightweight design and efficiency optimization.

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

# A. Appendix

### A.1. More Results Across Backbone.

Table 8 complements Table 1 by reporting BCP results with RN101 and ViT-B/32 backbones. BCP achieves consistent and substantial improvements under both architectures, indicating that its effectiveness generalizes across backbone families. These results suggest that BCP is robust to the choice of backbone.

*Table 8.* Comparison with CLIP and SOTAs on adapting multi-label instances with different architectures.

| | METHOD | BP-FREE | COCO2014 | COCO2017 | VOC2007 | VOC2012 | NUSWIDE | AVERAGE |
|---|---|---|---|---|---|---|---|---|
| RN-101 | CLIP (RADFORD ET AL., 2021) | √ | 48.83 | 48.15 | 76.72 | 74.21 | 41.93 | 57.97 |
| | DMN (ZHANG ET AL., 2024C) | √ | 46.28 | 45.44 | 76.82 | 75.32 | 42.71 | 57.31 |
| | TDA (KARMANOV ET AL., 2024) | √ | 50.19 | 49.78 | 78.12 | 77.13 | 43.13 | 59.67 |
| | TPT (SHU ET AL., 2022) | × | 49.71 | 48.89 | 74.82 | 73.39 | 43.10 | 57.98 |
| | DIFFTPT (FENG ET AL., 2023) | × | 49.45 | 49.19 | 74.98 | 74.31 | 42.93 | 58.17 |
| | RLCF (ZHAO ET AL., 2024) | × | 40.53 | 39.79 | 71.21 | 69.63 | 31.77 | 50.59 |
| | DOTA (HAN ET AL., 2025) | √ | 48.46 | 60.07 | 83.00 | 79.05 | 19.73 | 58.06 |
| | BEM (WU ET AL., 2025) | × | 52.92 | 52.24 | 78.72 | 78.13 | 43.62 | 61.13 |
| | **BCP** | √ | **62.84** | **62.66** | **87.33** | **86.44** | **50.84** | **70.02** |
| ViT-B/32 | CLIP (RADFORD ET AL., 2021) | √ | 50.31 | 50.15 | 77.18 | 76.85 | 42.90 | 59.48 |
| | DMN (ZHANG ET AL., 2024C) | √ | 49.32 | 48.13 | 77.42 | 76.60 | 43.41 | 58.98 |
| | TDA (KARMANOV ET AL., 2024) | √ | 51.23 | 51.49 | 77.62 | 77.12 | 44.13 | 60.32 |
| | TPT (SHU ET AL., 2022) | × | 48.12 | 48.63 | 74.21 | 71.93 | 43.63 | 57.30 |
| | DIFFTPT (FENG ET AL., 2023) | × | 48.73 | 49.19 | 74.50 | 72.98 | 43.42 | 57.76 |
| | RLCF (ZHAO ET AL., 2024) | × | 50.28 | 49.59 | 77.12 | 76.83 | 43.29 | 59.42 |
| | DOTA (HAN ET AL., 2025) | √ | 54.66 | 61.12 | 84.76 | 81.71 | 33.60 | 63.17 |
| | BEM (WU ET AL., 2025) | × | 52.83 | 52.99 | 78.70 | 77.97 | 44.12 | 61.32 |
| | **BCP** | √ | **62.97** | **62.14** | **87.2** | **85.76** | **52.71** | **70.15** |

### A.2. More Ablation Studies of Anchor Impact Across Architectures.

Table 9 complements Table 2 by reporting mAP on the confident anchor subset with $P_{\max} > \mu$, further split into *Anchor-Hit*, where the predicted anchor is a ground-truth positive, and *Anchor-Miss* otherwise. Across RN50, RN101, and ViT-B/32, BCP improves mAP primarily on *Anchor-Hit*, while it is slightly worse than frozen CLIP on *Anchor-Miss* on average. This pattern indicates that the correction is most effective when the anchor is reliable, and that incorrect conditioning can over constrain the update. The hit share is high on COCO and VOC but much lower on NUSWIDE, which helps explain why the overall benefit depends on how often confident anchors are correct.

### A.3. Ablations On Multiple Anchors.

We ablate the single anchor design by extending the correction to multiple anchors. For each test image, we compute the zero-shot posterior via Softmax and select the top $A$ labels as an anchor set. For every non-anchor label, we compute its PMI correction with respect to each anchor in the set, then average these $A$ PMI values to obtain one correction term, and finally add this term to the corresponding logit. We keep the logits of anchor labels unchanged, since they already serve as conditioning context.

Table 10 shows that increasing the number of anchors consistently degrades performance across datasets, with the best results at $A = 1$. This behavior is expected because naive aggregation mixes heterogeneous conditioning contexts. In multi label recognition, the top ranked labels often include fine grained variants or visually confusing concepts that are not jointly true. Averaging their PMI signals therefore introduces contradictory updates and shrinks the effective correction magnitude, while also amplifying estimation noise for anchors with lower base rates. Overall, these results support our choice of using a single high confidence anchor as the most stable and informative context for Bayesian prior correction.

*Table 9.* Impact of anchor correctness on performance (mAP) across architecture.

| | SETTING | METHOD | COCO2014 | COCO2017 | VOC2007 | VOC2012 | NUSWIDE | AVERAGE |
|---|---|---|---|---|---|---|---|---|
| RN50 | ANCHOR-HIT | CLIP | 55.33 | 55.17 | 81.69 | 80.62 | 68.60 | 68.28 |
| | ANCHOR-HIT | BCP | **67.59** | **67.74** | **90.91** | **89.49** | **81.40** | **79.43** |
| | ANCHOR-MISS | CLIP | 21.34 | 27.41 | **31.43** | **32.02** | 13.64 | **25.17** |
| | ANCHOR-MISS | BCP | **24.28** | **29.52** | 26.87 | 28.23 | **14.40** | 24.66 |
| | HIT SHARE (%) | - | 91.83 | 91.75 | 84.97 | 87.46 | 51.62 | 81.53 |
| | MISS SHARE (%) | - | 8.17 | 8.25 | 15.03 | 12.54 | 48.38 | 18.47 |
| RN101 | ANCHOR-HIT | CLIP | 57.05 | 57.11 | 83.55 | 82.20 | 67.79 | 69.54 |
| | ANCHOR-HIT | BCP | **67.44** | **68.11** | **91.05** | **89.57** | **81.90** | **79.61** |
| | ANCHOR-MISS | CLIP | 21.45 | 26.12 | **31.61** | **28.11** | 12.94 | **24.05** |
| | ANCHOR-MISS | BCP | **23.75** | **31.31** | 21.46 | 21.45 | **13.31** | 22.26 |
| | HIT SHARE (%) | - | 91.50 | 91.96 | 89.18 | 91.09 | 51.13 | 82.97 |
| | MISS SHARE (%) | - | 8.50 | 8.04 | 10.82 | 8.91 | 48.87 | 17.03 |
| ViT-B/32 | ANCHOR-HIT | CLIP | 59.18 | 59.78 | 81.04 | 80.90 | 69.01 | 69.98 |
| | ANCHOR-HIT | BCP | **67.99** | **68.02** | **91.01** | **88.96** | **81.88** | **79.57** |
| | ANCHOR-MISS | CLIP | 21.58 | 25.68 | **30.92** | **29.16** | 12.78 | **24.02** |
| | ANCHOR-MISS | BCP | **22.40** | **27.01** | 26.34 | 25.34 | **13.43** | 22.90 |
| | HIT SHARE (%) | - | 93.42 | 93.18 | 91.75 | 93.52 | 57.69 | 85.91 |
| | MISS SHARE (%) | - | 6.58 | 6.82 | 8.25 | 6.48 | 42.31 | 14.09 |

*Table 10.* Ablations on multiple anchors.

| Anchor count | COCO2014 | COCO2017 | VOC2007 | VOC2012 | NUSWIDE | Average |
|---|---|---|---|---|---|---|
| 1 | 65.47 | 64.92 | 88.41 | 87.37 | 52.78 | 71.79 |
| 2 | 62.47 | 61.58 | 88.15 | 87.01 | 50.47 | 69.93 |
| 3 | 59.70 | 58.26 | 87.43 | 86.27 | 48.06 | 67.94 |

## A.4. Performance on the Last 50% of Test Samples.

To evaluate long horizon behavior in multi label test time adaptation, we report mAP on the second half of each test stream, namely the last 50% of samples, on COCO2014, COCO2017, VOC2007, VOC2012, and NUSWIDE. This setting reflects the stage after many online updates, where estimation noise and model drift can accumulate, so performance on later samples directly measures stability under sustained distribution shift.

Table 11 compares BCP with the zero shot CLIP baseline on this subset across four backbones. BCP improves mAP on every dataset and every architecture, with average gains of **+11.14** on RN50, **+11.04** on RN101, **+10.05** on ViT-B/32, and **+8.21** on ViT-B/16. At the dataset level, gains are consistently positive and range from **+7.26** to **+13.00** mAP across all architecture and dataset pairs.

## A.5. Robustness Under Non-i.i.d. Test Streams.

The main experiments follow the standard randomly shuffled online evaluation protocol. To further evaluate robustness under more realistic test streams, we test BCP under non-i.i.d. distribution settings with three severity levels controlled by $\gamma$. Smaller $\gamma$ corresponds to a stronger non-i.i.d. stream, where samples are more locally concentrated by distributional structure rather than uniformly shuffled.

As shown in Table 12, BCP degrades only mildly as the stream becomes more non-i.i.d. The average mAP changes from 69.15 under low severity to 68.66 under high severity, corresponding to a drop of only 0.49. These results suggest that the online conditional-prior estimator remains stable under the tested realistic stream protocol, rather than relying on a special test-sample ordering.

*Table 11.* Performance comparison for the last 50% samples.

| ARCH | METHOD | COCO2014 | COCO2017 | VOC2007 | VOC2012 | NUSWIDE | AVERAGE |
|------|--------|----------|----------|---------|---------|---------|---------|
| RN50 | CLIP | 47.46 | 47.85 | 75.31 | 74.13 | 41.40 | 57.23 |
| | **BCP** | **60.46** | **60.14** | **86.73** | **84.96** | **49.55** | **68.37** |
| RN101 | CLIP | 48.90 | 47.90 | 76.64 | 75.95 | 42.17 | 58.31 |
| | **BCP** | **61.38** | **59.99** | **87.73** | **86.46** | **51.21** | **69.35** |
| VIT-B/32 | CLIP | 50.32 | 50.05 | 77.36 | 76.69 | 43.59 | 59.60 |
| | **BCP** | **61.30** | **61.51** | **87.13** | **85.67** | **52.62** | **69.65** |
| VIT-B/16 | CLIP | 54.49 | 54.35 | 79.37 | 79.02 | 45.87 | 62.62 |
| | **BCP** | **63.52** | **62.72** | **87.85** | **86.92** | **53.13** | **70.83** |

*Table 12.* BCP performance under non-i.i.d. test streams on ViT-B/16.

| SETTING | COCO14 | COCO17 | VOC07 | VOC12 | NUS | AVG. |
|---------|--------|--------|-------|-------|-----|------|
| LOW ($\gamma = 0.1$) | 62.75 | 64.73 | 86.00 | 85.63 | 46.66 | 69.15 |
| MEDIUM ($\gamma = 0.01$) | 62.87 | 62.37 | 86.11 | 85.23 | 46.99 | 68.71 |
| HIGH ($\gamma = 0.001$) | 62.78 | 62.20 | 85.89 | 85.43 | 46.98 | 68.66 |

## A.6. Paramrter Sensitivity Analysis.

We evaluate hyperparameter robustness by studying the sensitivity to the confidence threshold $\mu$ that controls which samples are treated as confident anchors for prior correction. We sweep $\mu$ over 0.1, 0.3, 0.5, 0.75, and 0.95, and report mAP on COCO2014, COCO2017, NUSWIDE, VOC2007, and VOC2012 as shown in Figure **??**. :contentReference[oaicite:0]index=0 :contentReference[oaicite:1]index=1 Performance varies smoothly with $\mu$ and stays close to the peak across the full sweep. The best results consistently occur near $\mu = 0.5$, while more permissive or stricter thresholds lead to only modest degradation. These results indicate that BCP does not depend on a narrow hyperparameter setting, and $\mu = 0.5$ is a reliable default.

## A.7. Warmup Variant for Early-stage Stability.

At very early timesteps, the running co-occurrence matrix $\mathbf{U}^t$ and marginal vector $\mathbf{m}^t$ can have high variance because they are estimated from only a small number of samples. To reduce the effect of noisy early statistics, we evaluate a conservative warmup variant of BCP that delays the Bayesian correction until the normalized running statistics become sufficiently stable. Importantly, this variant does not change the correction rule itself; it only postpones when Eq. (9) is activated.

Specifically, we monitor the average element-wise change of the normalized co-occurrence matrix and marginal vector:

$$\Delta_U^{(t)} = \frac{1}{K^2} \sum_{i=1}^{K} \sum_{j=1}^{K} \left| \frac{U_{i,j}^t}{t} - \frac{U_{i,j}^{t-1}}{t-1} \right|, \tag{16}$$

$$\Delta_m^{(t)} = \frac{1}{K} \sum_{i=1}^{K} \left| \frac{m_i^t}{t} - \frac{m_i^{t-1}}{t-1} \right|. \tag{17}$$

BCP remains in the warmup phase until both $\Delta_U^{(t)} < 10^{-4}$ and $\Delta_m^{(t)} < 10^{-4}$ are satisfied. During warmup, we simply use the zero-shot prediction, i.e., $\mathbf{s}^{*(t)} = \mathbf{s}^{(t)}$; after the stability criteria are met, we apply the same Bayesian correction as the original BCP.

Table 13 shows that the warmup variant brings a consistent but modest improvement across benchmarks. This confirms that early-stage statistic noise is a valid concern, and that it can be mitigated by a conservative activation strategy without modifying the core BCP update.

## A.8. Conditional Prior Visualization.

Figure 5 visualizes the class conditional priors estimated by BCP after test time adaptation on VOC2007, VOC2012, COCO2014, COCO2017, and NUSWIDE. For label indices $a$ and $b$, each heatmap entry reports the estimated conditional

*Table 13.* Comparison of BCP and its warmup variant on ViT-B/16.

| METHOD | COCO14 | COCO17 | VOC07 | VOC12 | NUS | AVG. |
|---|---|---|---|---|---|---|
| BCP | 65.47 | 64.92 | 88.41 | 87.37 | 52.78 | 71.79 |
| **BCP-WARMUP** | **65.72** | **65.08** | **88.53** | **87.60** | **52.83** | **71.95** |

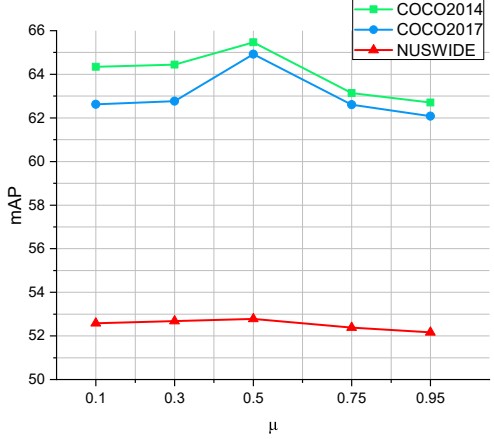 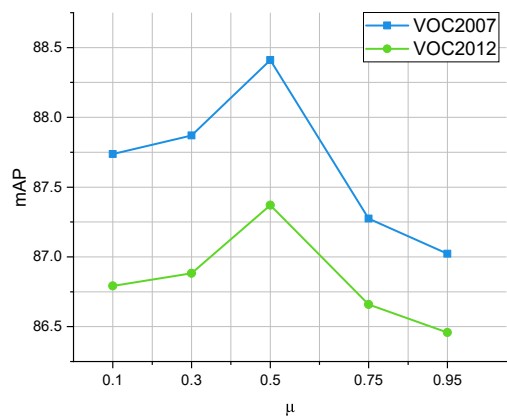

*Figure 4.* Sensitivity analysis of the hyperparameter $\mu$.

dependency $\hat{P}\{y_b = 1 \mid y_a = 1\}$, denoted as $P_{b|a}$ in the figure. We set diagonal entries to zero to highlight cross class relationships.

We highlight three observations. First, COCO2014 and COCO2017 show clear block structure that follows semantic relatedness. Kitchen and dining categories form a coherent cluster, for example fork, knife, spoon, bowl, cup, and wine glass tend to mutually reinforce each other. Transportation labels also cluster, including bicycle, motorcycle, car, bus, truck, and train. Similar structure appears in sports and outdoor equipment, including skis, snowboard, skateboard, surfboard, and sports ball. These blocks indicate that the adapted priors capture stable co occurrence regularities instead of diffuse correlations, which is essential for using them as a corrective signal.

Second, datasets with similar label spaces produce similar dependency profiles, most notably COCO2014 versus COCO2017 and VOC2007 versus VOC2012. This consistency suggests that the online estimates are not dominated by sampling noise. At the same time, NUSWIDE exhibits a more diffuse pattern, which is consistent with its broader vocabulary and weaker annotation consistency. This contrast supports our motivation that dependency structure is dataset dependent and can shift in ways that require adaptation.

Third, the heatmaps are generally asymmetric because $\hat{P}\{y_b = 1 \mid y_a = 1\}$ can differ from $\hat{P}\{y_a = 1 \mid y_b = 1\}$. This asymmetry often reflects differences in base rates and specificity. For example, conditioning on a specific accessory or sports item tends to raise the probability of person more than conditioning on person raises the probability of that item. This directionality directly supports our anchor conditioned correction, where a reliable predicted anchor provides informative context for refining the remaining logits.

Overall, these visualizations support our method design. Multi label distribution shift can change cross label dependencies in a dataset dependent manner, and adapting class conditional priors provides an explicit and lightweight way to track these changes during test time inference without modifying the backbone.

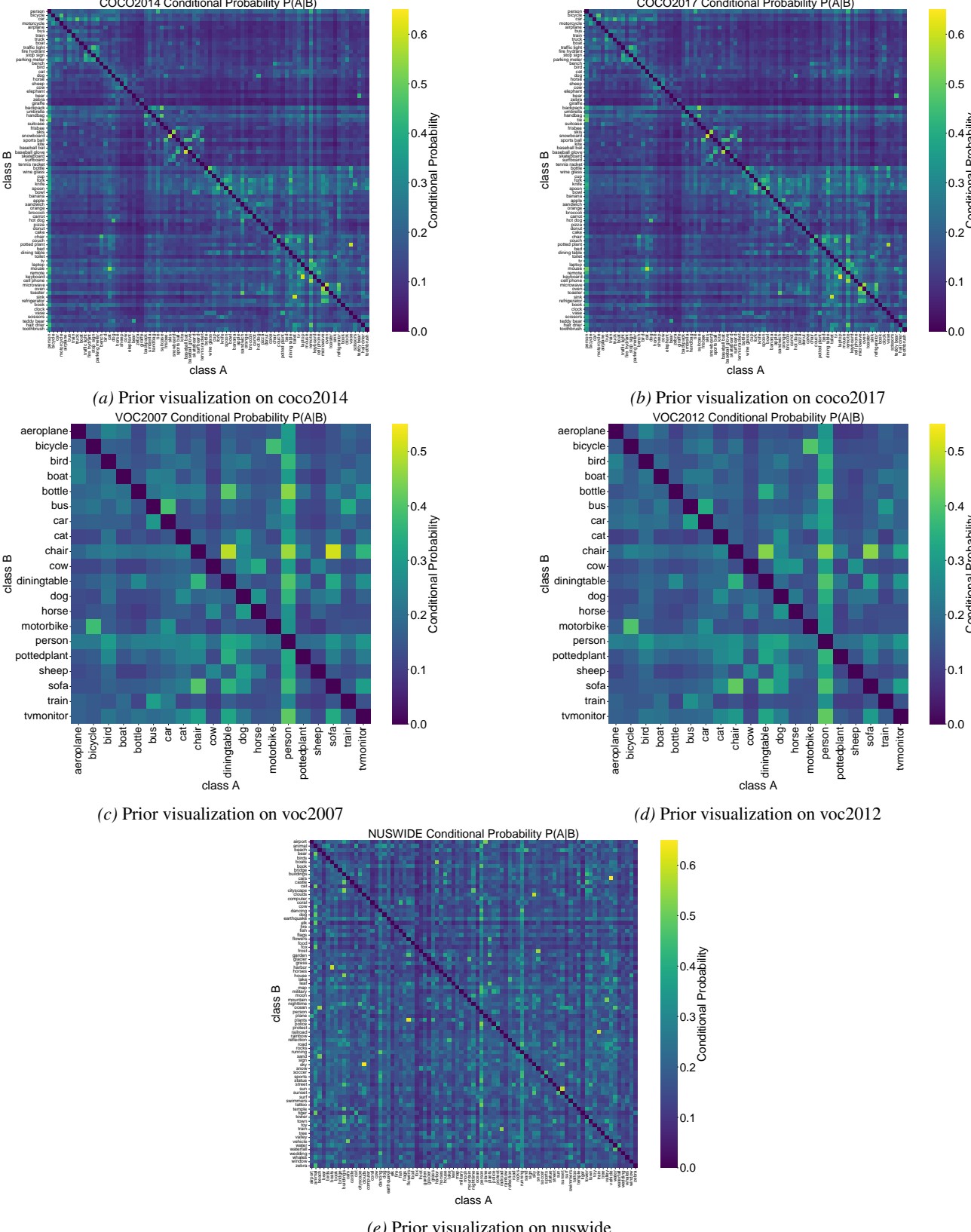

*(a)* Prior visualization on coco2014

*(b)* Prior visualization on coco2017

*(c)* Prior visualization on voc2007

*(d)* Prior visualization on voc2012

*(e)* Prior visualization on nuswide

*Figure 5.* Prior visualization on five benchmarks.

