# OpenReview forum: "Multi-Label Test-Time Adaptation with Bayesian Conditional Priors"
_ICML.cc/2026/Conference — ICML 2026 regular_

### Official Review · Reviewer_fmtz · 2026-03-02

**Soundness:** 3
**Presentation:** 3
**Significance:** 3
**Originality:** 3
**Overall Recommendation:** 5
**Confidence:** 5

**Summary:**

Summary
The authors propose Bayesian Conditional Priors (BCP). It treats CLIP zero-shot outputs as visual likelihood times marginal priors, then estimates label co-occurrence from the test stream to compute conditional priors using high-confidence anchors. The final logits are corrected with a closed-form PMI-based solution, adding label dependency without updating model parameters.

**Compliance With Llm Reviewing Policy:**

Affirmed.

**Final Justification:**

I raise my score and confidence to support acceptance of this paper.

**Key Questions For Authors:**

See weakness.

**Limitations:**

See weakness.

**Strengths And Weaknesses:**

Strength
1. The paper is well written and easy to follow.
2. The formulation that the logit correction term is equivalent to pointwise mutual information (PMI) is elegant. It turns a complex adaptation problem into a lightweight statistical calibration step.
3. Extensive experiments have shown the significant performance gain of BCP on COCO, VOC, and NUSWIDE.
4. BCP avoids back-propagation and complex caption retrieval. Its inference latency is less than half of the SOTA method BEM.

Weakness
1. The core assumption is that the Top-1 prediction is reliable. However, If the model is highly confident about a non-existent object, the PMI correction may boost labels that co-occur with this incorrect anchor, causing error propagation.
2. BCP depends on historical statistics. At the beginning of the test stream, statistical estimates have high variance, and corrections may be unstable or harmful. In realistic streaming scenarios, accumulated global statistics may drift due to the non-iid test stream. For example, under the setting described in [e], BCP is likely to fail.
3. Missing comparision with recent CLIP-based TTA works [a-d].

\[a] Backpropagation-Free Test-Time Adaptation via Probabilistic Gaussian Alignment\
\[b] Statistics Caching Test-Time Adaptation for Vision-Language Models\
\[c] STS: Test-Time Spectrum-Aware Latent Steering for Zero-Shot Generalization in Vision-Language Models\
\[d] Training-Free Test-Time Adaptation via Shape and Style Guidance for Vision-Language Models\
\[e] Realistic test-time adaptation of vision-language models.

---

> ### Author Rebuttal · Authors · 2026-03-31
>
> We thank the reviewer for the valuable feedback and insightful comments. Below we address the concerns and provide further clarification and supporting experimental results.
>
> **`Response to Weakness 1:`**
> Due to the space limitations of the rebuttal, we kindly refer the reviewer to our response to reviewer tzvE’s Question 1 for a detailed discussion, and we appreciate the reviewer’s understanding.
>
> **`Response to Weakness 2:`**
> **(1) Early-stage instability.** We agree that at very early timesteps, the running estimates can have high variance, which may make the Bayesian correction unstable. To address this issue, we additionally introduce a warmup variant of BCP that delays the correction until the running statistics become sufficiently stable.
>
> Recall that $\mathbf{U}^{t}$ and $\mathbf{m}^{t}$ in Eq.~(7) are running sums. We monitor the stability of their normalized forms, $\mathbf{U}^{t}/t$ and $\mathbf{m}^{t}/t$, by measuring the average element-wise change between two consecutive steps:
>
> $$
> \Delta_U^{(t)} = \frac{1}{K^2} \sum_{i=1}^{K} \sum_{j=1}^{K} \left| \frac{U_{i,j}^{t}}{t} - \frac{U_{i,j}^{t-1}}{t-1} \right|,
> $$
>
> $$
> \Delta_m^{(t)} = \frac{1}{K} \sum_{i=1}^{K} \left| \frac{m_i^{t}}{t} - \frac{m_i^{t-1}}{t-1} \right|.
> $$
>
> We keep BCP in a warmup phase until both $\Delta_U^{(t)} < 10^{-4}$ and $\Delta_m^{(t)} < 10^{-4}$ are satisfied. During warmup, we simply use the zero-shot prediction, i.e.,
>
> $$
> \mathbf{s}^{*(t)}=\mathbf{s}^{(t)},
> $$
>
> and only activate the Bayesian correction after the running statistics become sufficiently stable. In other words, the warmup version does **not** change the core BCP update; it only introduces a conservative activation rule to avoid noisy early corrections.
>
> Empirically, as shown in Table 2, the **warmup variant** brings a further slight improvement over the BCP across benchmarks, suggesting that the reviewer’s concern is valid and can be effectively mitigated by a cautious activation strategy.
>
> **(2) Non-i.i.d. / realistic test stream.** We also agree that robustness under non-i.i.d. test streams is important. To directly address this concern, we further evaluated BCP under the **realistic setting** of [e]. As shown in table 1, the average mAP is 69.15 / 68.71 / 68.66 under low / medium / high non-i.i.d. severity, respectively, i.e., only a 0.49 drop from low to high severity. This suggests that, under the tested realistic setting, BCP degrades only mildly rather than failing outright.
>
> **Table 1.** BCP performance under three non-i.i.d. distribution settings on COCO2014, COCO2017, VOC2007, VOC2012, and NUSWIDE.
> | Setting | COCO2014 | COCO2017 | VOC2007 | VOC2012 | NUSWIDE | AVERAGE |
> | :---: | :---: | :---: | :---: | :---: | :---: | :---: |
> | Low ($\gamma$=0.1) | 62.75 | 64.73 | 86.00 | 85.63 | 46.66 | 69.15 |
> | Medium ($\gamma$=0.01) | 62.87 | 62.37 | 86.11 | 85.23 | 46.99 | 68.71 |
> | High ($\gamma$=0.001) | 62.78 | 62.20 | 85.89 | 85.43 | 46.98 | 68.66 |
>
> **`Response to Weakness 3:`**
> We have added a comparison with recent CLIP-based TTA methods [a--d] in table 2. For fairness, all methods are evaluated with the same CLIP ViT-B/16 backbone, using the same evaluation setting, and we use the authors' default / best reported hyperparameters whenever available. Methods marked with **\*** were reproduced by us because their official code is unavailable.
>
> Under this setting, BCP already achieves the best average mAP (71.79 vs.\ 69.24), and the warmup version further improves to 71.95, achieving the best result on all five benchmarks.
>
> **Table 2.** Comparison with recent CLIP-based TTA works.
> |  | Method | COCO2014 | COCO2017 | VOC2007 | VOC2012 | NUSWIDE | Average |
> | :---: | :---: | :---: | :---: | :---: | :---: | :---: | :---: |
> | ViT-B/16 | STS*[c] | 59.80 | 59.09 | 85.09 | 83.73 | 47.23 | 66.99 |
> | ViT-B/16 | ADAPT[a] | 61.91 | 59.84 | 86.34 | 84.92 | 43.18 | 67.24 |
> | ViT-B/16 | SSG*[d] | 61.49 | 60.96 | 86.02 | 84.53 | 47.93 | 68.19 |
> | ViT-B/16 | SCA[b] | 63.31 | 61.81 | 86.99 | 86.00 | 48.08 | 69.24 |
> | ViT-B/16 | BCP | 65.47 | 64.92 | 88.41 | 87.37 | 52.78 | 71.79 |
> | ViT-B/16 | **BCP with warmup** | **65.72** | **65.08** | **88.53** | **87.60** | **52.83** | **71.95** |
>
> **`Summary:`**
> We sincerely thank the reviewer again for the thoughtful, constructive, and insightful comments. In response, we have provided additional clarifications, introduced a warmup variant to address the concern on early-stage instability, and added further experiments under realistic non-i.i.d. test streams as well as comparisons with recent CLIP-based TTA methods. These results consistently support the effectiveness and robustness of our method, while also confirming that the reviewer’s concerns are valid and can be properly addressed. We will incorporate these clarifications, analyses, and experimental results into the final version to make the paper clearer and more complete. We also remain happy to further clarify any remaining questions the reviewer may have.

---

> > ### Author Rebuttal · Reviewer_fmtz · 2026-04-01
> >
> > I appreciate the authors' effort to address my questions in a limited time. All of my concerns have been clarified. I raise my score and confidence to support acceptance of this paper. I think parts of the rebuttal can be added to the final version to further improve the paper, if accepted. Good luck.

---

> > > ### Author Response · Authors · 2026-04-01
> > >
> > > Thank you very much for your thoughtful feedback and support. We sincerely appreciate your time and recognition of our rebuttal. We also thank you for the helpful suggestion, and we will add relevant parts of the rebuttal to the final version to further improve the paper.

---

### Official Review · Reviewer_PCds · 2026-03-08

**Soundness:** 3
**Presentation:** 3
**Significance:** 3
**Originality:** 3
**Overall Recommendation:** 4
**Confidence:** 4

**Summary:**

This paper proposes a test-time adaptation for multi-label image classification. The proposed method uses a Bayesian perspective and takes into account of the correlation between labels, which is something previous TTA methods usually do not consider.

**Compliance With Llm Reviewing Policy:**

Affirmed.

**Final Justification:**

I would keep my positive rating

**Key Questions For Authors:**

Please refer to "weakness" above

**Limitations:**

yes

**Strengths And Weaknesses:**

Strength
=======

1) The idea of taking into account of the correlation of labels in TTA is interesting. As far as I know, it is also novel.
2) The paper proposes a theoretical connection between TTA and Bayesian conditional prior. The proposed TTA method is based on some principled Bayesian formulation instead of some ad-hoc heuristics.
3) The experiment results show a fairly significant improvement over previous methods

Weakness
========
1) The proposed method implicitly assumes a particular ordering of test examples. And the result depends on this ordering. It is not clear to me what ordering is used to get the result in the experiment, and how sensitive the result is to this ordering.
2) In Algorithm 1, when "t" is small (i.e. for those test examples in the beginning), the statistics computed via Eq 8 will probably be unreliable, since you have not see enough examples yet. So it seems the results may be imbalanced depending on whether you at the start of the end in the streaming of test examples

---

> ### Author Rebuttal · Authors · 2026-03-31
>
> We thank the reviewer for the thoughtful comments and constructive suggestions. Below, we address the concerns and provide additional clarification and empirical results.
>
> **`Response to Weakness 1:`**
> **BCP** is an **online method** and therefore operates on a **test stream**, but it follows the same evaluation setting as other **test-time adaptation (TTA)** methods and does not rely on any special ordering of test samples. In the main experiments, we use one random permutation of the test stream. To examine sensitivity to ordering, we additionally repeated the evaluation with **five different random permutations** generated by different seeds. As shown in **Table 1**, the results are highly consistent across different random seeds, with a standard deviation of only **0.04** in the **cross-dataset average performance**. This indicates that **BCP is robust to the ordering of the test stream** under randomly shuffled evaluation.
>
> **Table 1.** Results under five different random orderings of the test stream (**ViT-B/16**, **BCP**).
> | Dataset  | Seed1 | Seed2 | Seed3 | Seed4 | Seed5 | Average |
> | :---: | :---: | :---: | :---: | :---: | :---: | :---: |
> | COCO2014 | 65.48 | 65.40 | 65.48 | 65.49 | 65.42 | 65.45 |
> | COCO2017 | 64.85 | 64.79 | 64.93 | 64.85 | 64.76 | 64.84 |
> | VOC2007  | 88.31 | 88.21 | 88.31 | 88.19 | 88.24 | 88.25 |
> | VOC2012  | 87.38 | 87.42 | 87.41 | 87.29 | 87.36 | 87.37 |
> | NUSWIDE  | 52.65 | 52.71 | 52.89 | 52.92 | 52.74 | 52.78 |
> | AVERAGE  | 71.73 | 71.71 | 71.80 | 71.75 | 71.70 | 71.74 |
>
> **`Response to Weakness 2:`**
> We agree with the reviewer that the **running statistics** in Eq.~(8) are noisier at **early timesteps** because they are estimated from a limited history. In the current method, this effect is already partially controlled by two design:
> (i) BCP updates the statistics only using the zero-shot CLIP posterior.
> (ii) BCP applies the correction only when the anchor confidence of a sample exceeds the threshold $\mu$.
>
> To further reduce early-stage noise, we additionally implemented a **warmup variant of BCP**. The key idea is simple: we delay activating the same **Bayesian correction** until the running statistics become sufficiently stable. Importantly, this does **not** change the correction rule itself; it only postpones when Eq.~(9) is activated.
>
> Specifically, let $\mathbf{U}^{t}$ and $\mathbf{m}^{t}$ denote the **running co-occurrence matrix** and **marginal vector** in Eq.~(7). We monitor the average element-wise change of their normalized forms:
>
> $$
> \Delta_U^{(t)} = \frac{1}{K^2} \sum_{i=1}^{K} \sum_{j=1}^{K} \left| \frac{U_{i,j}^{t}}{t} - \frac{U_{i,j}^{t-1}}{t-1} \right|,
> $$
>
> and
>
> $$
> \Delta_m^{(t)} = \frac{1}{K} \sum_{i=1}^{K} \left| \frac{m_i^{t}}{t} - \frac{m_i^{t-1}}{t-1} \right|.
> $$
>
> We keep the model in a **warmup phase** until both $\Delta_U^{(t)} < 10^{-4}$ and $\Delta_m^{(t)} < 10^{-4}$. During warmup, we simply use the **zero-shot prediction**, i.e., $\mathbf{s}^{*(t)}=\mathbf{s}^{(t)}$. Once both criteria are satisfied, we start applying the same correction as in Eq.~(9).
>
> Empirically, this **warmup variant** brings a further slight improvement over **BCP** across benchmarks, as shown in **Table 2**. Therefore, while the reviewer's concern about **early-step statistics** is valid, the effect can be effectively mitigated by a **conservative activation strategy**.
>
> **Table 2.** Comparison between BCP and BCP with warmup on the ViT-B/16 backbone.
> | Method | COCO2014 | COCO2017 | VOC2007 | VOC2012 | NUSWIDE | Average |
> | :---: | :---: | :---: | :---: | :---: | :---: | :---: |
> | BCP | 65.47 | 64.92 | 88.41 | 87.37 | 52.78 | 71.79 |
> | **BCP with warmup** | **65.72** | **65.08** | **88.53** | **87.60** | **52.83** | **71.95** |
>
> **`Summary:`**
> We sincerely thank the reviewer again for the thoughtful and constructive feedback. In response, we have provided additional clarification and empirical evidence regarding both concerns. Specifically, we verified that BCP is robust to different random orderings of the test stream, and we further introduced a **simple warmup variant** to mitigate the noise of **early-stage running statistics**, which yields a further slight improvement over the BCP across benchmarks. We will incorporate these clarifications, analyses, and additional experimental results into the final version. We will also further streamline the presentation to make the method and its motivation clearer. We hope these responses address the reviewer's concerns, and we would be grateful for any further questions or suggestions.

---

> > ### Author Rebuttal · Reviewer_PCds · 2026-04-01
> >
> > The rebuttal addressed my concern.

---

> > > ### Author Response · Authors · 2026-04-01
> > >
> > > Thank you for your update and for carefully considering our rebuttal. We appreciate your time and are glad that it addressed your concern.

---

### Official Review · Reviewer_tzvE · 2026-03-10

**Soundness:** 3
**Presentation:** 3
**Significance:** 3
**Originality:** 3
**Overall Recommendation:** 4
**Confidence:** 4

**Summary:**

The paper addresses the limitation of existing multi-label Test-Time Adaptation (TTA) methods, which often ignore label co-occurrence structures. To resolve this, the authors introduce Bayesian Conditional Priors (BCP) Estimation, a gradient-free TTA approach that injects label dependencies without fine-tuning the VLM backbone. By selecting a high-confidence anchor label, the method refines zero-shot logits using an online estimation of conditional priors, which mathematically aligns with a pointwise mutual information update. Extensive experiments across standard multi-label benchmarks and multiple CLIP backbones demonstrate that BCP consistently outperforms strong TTA baselines, eliminating the need for backpropagation while achieving higher accuracy and lower latency.

**Compliance With Llm Reviewing Policy:**

Affirmed.

**Key Questions For Authors:**

1. Have you considered implementing a semantic-distance checking mechanism or a dynamic confidence threshold that could detect an unreliable anchor and scale down the PMI correction to prevent downstream error propagation?

2. In datasets with highly imbalanced or long-tailed label distributions, does the BCP framework systematically suppress rare labels that do not frequently co-occur with the dominant anchor?

3. If multiple non-anchor labels share nearly identical co-occurrence statistics with the chosen anchor, does the framework rely entirely on the original zero-shot visual evidence to rank them?

**Limitations:**

yes

**Strengths And Weaknesses:**

Strengths:

The paper is clearly structured and well-written. The motivation is straightforward. The mathematical formulation is logically sound. Viewing zero-shot logits as proxies for marginal posteriors and adjusting them via a closed-form PMI correction is a principled way to inject correlation priors without disrupting pre-trained feature alignments. BCP acts as a plug-and-play module that bypasses iterative gradient updates. This dramatically reduces latency and memory costs, making it highly suitable for real-time deployment.

Weaknesses:

1. While Appendix A.3 adequately explains why a single anchor is preferred over multiple anchors, relying exclusively on the top-1 prediction introduces a bottleneck. If the model predicts a completely incorrect top-1 anchor, this false signal propagates errors to the remaining label distributions.

2. The authors candidly discuss the "Anchor-Miss" scenario in Section 4.2, noting that performance drops on datasets like VOC because incorrect anchors often come from entirely different semantic scenes. However, there is no dynamic mechanism to detect when a predicted anchor is semantically isolated or unreliable in order to attenuate the PMI correction.

3. It is not entirely clear how the anchor-conditioned prior affects long-tail or rare classes. If a dominant anchor is selected, the PMI correction might systematically suppress weaker but valid long-tail labels simply because they lack sufficient co-occurrence statistics in the online test stream.

---

> ### Author Rebuttal · Authors · 2026-03-31
>
> We thank the reviewer for the careful reading and constructive feedback. Below we address the concerns and provide additional clarification and empirical evidence.
>
> **`Response to Weakness 1, 2, and Question 1:`**
> First, BCP does not create a **cross-sample self-reinforcing error accumulation** loop through PMI correction. The running marginal and co-occurrence statistics are updated only from the **zero-shot CLIP posterior**, not the PMI-corrected logits. Therefore, an incorrect correction on one sample is **not fed back** into the online statistics and cannot amplify over future samples. The main risk is **intra-sample miscorrection** when the selected anchor is unreliable.
>
> **(1) On semantic-distance checking**
> We agree that semantic-distance based reliability checking is an interesting direction. In the current design, we apply PMI correction only when the anchor confidence exceeds a threshold $\mu$, which already filters out many unreliable anchors. Moreover, our Anchor-Miss analysis suggests that incorrect anchors are not uniformly harmful: when the predicted anchor remains semantically close to the true label set, PMI correction can still improve the ranking of relevant labels.
>
> At the same time, **semantic similarity is not equivalent to empirical co-occurrence structure**. BCP corrects logits using online conditional priors estimated from the target stream, while a semantic-distance module would introduce an external notion of similarity that may not align with the actual co-occurrence statistics under distribution shift. We therefore view semantic-distance checking as a promising extension rather than a necessary component, and we will add this discussion to the revised paper as future work.
>
> **(2) On dynamic confidence thresholds**
> We also explored adaptive thresholding strategies beyond fixed $\mu$. In particular, we tested:
>
> - **Class-specific threshold:** using the running mean anchor confidence for each class;
> - **Running-median threshold:**
> $$
> \mu_t = \mathrm{median}\left( \{\max p^{(k)}(\cdot \mid x)\}_{k=1}^{t} \right).
> $$
>
> However, neither outperformed the **fixed threshold $\mu=0.5$** in our experiments. For example:
>
> **Table 1** Performance under Different Thresholding Strategies
> | Dataset | Fixed $\mu=0.5$ | Class-specific mean | Running median |
> |:---:|:---:|:---:|:---:|
> | VOC2012 | 87.37 | 86.52 | 86.02 |
> | COCO2017 | 64.92 | 64.05 | 63.93 |
>
> These results are also consistent with Appendix A.5, where BCP remains stable across a broad range of fixed $\mu$. This suggests that, under the current random test-stream setting, a fixed threshold already provides sufficiently robust gating.
>
> **`Response to Weakness 3 and Question 2:`**
> To examine whether BCP suppresses rare labels under imbalanced distributions, we split the categories of each dataset into **head / medium / tail** groups according to label frequency and report the mAP of each group. As shown in Table 2, BCP consistently outperforms both zsCLIP and SCA across all three groups on all five benchmarks, including the tail split. This does not support the concern that BCP systematically suppresses rare labels; instead, it suggests that the **conditional-prior correction** remains effective for low-frequency classes as well. Here, SCA denotes *Statistics Caching Test-Time Adaptation for Vision-Language Models*, a strong baseline added following the reviewer's suggestion.
>
> **Table 2.** Performance comparison under head-medium-tail splits
> | Dataset | Method | Head | Medium | Tail |
> |:---:|:---:|:---:|:---:|:---:|
> |  | zsCLIP | 42.47 | 62.53 | 57.93 |
> | COCO14 | SCA | 48.03 | 74.08 | 67.61 |
> |  | BCP | **51.84** | **75.51** | **69.22** |
> |  | zsCLIP | 42.29 | 60.45 | 59.51 |
> | COCO17 | SCA | 47.17 | 70.10 | 68.42 |
> |  | BCP | **51.42** | **73.08** | **70.29** |
> |  | zsCLIP | 80.15 | 70.61 | 89.30 |
> | VOC07 | SCA | 86.33 | 81.53 | 94.12 |
> |  | BCP | **88.13** | **82.56** | **95.57** |
> |  | zsCLIP | 80.42 | 71.78 | 87.50 |
> | VOC12 | SCA | 86.04 | 80.51 | 92.38 |
> |  | BCP | **87.58** | **81.60** | **93.80** |
> |  | zsCLIP | 47.57 | 43.15 | 46.01 |
> | NUSWIDE | SCA | 48.20 | 47.66 | 48.37 |
> |  | BCP | **52.47** | **47.87** | **55.90** |
>
> **`Response to Question 3:`**
> When multiple non-anchor labels have nearly identical co-occurrence statistics with the anchor, their correction terms also become similar, so their relative ranking is mainly determined by the zero-shot visual evidence. This follows our design principle: BCP calibrates the prior while preserving CLIP's **frozen visual discrimination**, rather than forcing extra ranking from weak statistics.
>
> **`Summary:`**
> In summary, BCP does not induce cross-sample error accumulation. Additional analysis shows that it remains robust in many anchor-error cases and does not systematically suppress rare labels. We also clarify its intended behavior when co-occurrence statistics are similar. We will incorporate these clarifications and new results into the final version.

---

> > ### Author Rebuttal · Reviewer_tzvE · 2026-04-03
> >
> > The authors' response and supplementary experiments have addressed most of my concerns; therefore, I have decided to maintain a positive evaluation and will not change my score.

---

> > > ### Author Response · Authors · 2026-04-03
> > >
> > > Dear reviewer tzvE:
> > >
> > > Thank you sincerely for taking the time to review our rebuttal and for thoughtfully considering our clarifications. We truly appreciate your positive score and your constructive comments, which help us strengthen the quality and clarity of our work.
> > >
> > > Best regards,
> > >
> > > The Authors

---

### Official Review · Reviewer_J3X4 · 2026-04-03

**Soundness:** 2
**Presentation:** 2
**Significance:** 2
**Originality:** 2
**Overall Recommendation:** 3
**Confidence:** 2

**Summary:**

This paper proposes a gradient-free test-time adaptation method, which injects label dependency without tuning the backbone. The authors select anchor labels to help the Bayesian refinement in the training-free manner. The experiments are extensive and helpful to validate the effectiveness of the model.

**Compliance With Llm Reviewing Policy:**

Affirmed.

**Final Justification:**

After reading the rebuttal and the comments from the other reviewers, I decided to maintain my score and believe the paper is below the bar of ICML.

**Key Questions For Authors:**

See weakness.

**Limitations:**

yes

**Strengths And Weaknesses:**

Pros.
- The problem studied in this paper is both interesting and significant.
- The paper is clearly written and well-structured.

Cons.
- BayesianTTA has incorporated TTA with training-free VLM inference. The novelty of the proposed method is limited. Moreover, the use of anchors/prototypes seems to be very common in training-free TTA. [1,2,3]
- The proof is not formal, which makes the paper below the bar of ICML. Please show the full theorem and a clear proof.
- Some of the baselines are not included, such as BayesianTTA, which should be carefully compared.
- The unique challenge in multi-labeled TTA is not clear. BayesianTTA can be directly applied to the problem with modest adaptation.

[1] Chakrabarty G, Sreenivas M, Biswas S. Santa: Source anchoring network and target alignment for continual test time adaptation[J]. Transactions on Machine Learning Research, 2023.

[2] Su Y, Xu X, Jia K. Revisiting realistic test-time training: Sequential inference and adaptation by anchored clustering[J]. Advances in Neural Information Processing Systems, 2022, 35: 17543-17555.

[3] Mishra S, Silva-Rodriguez J, Ayed I B, et al. Semantic Anchor Transport: Robust Test-Time Adaptation for Vision-Language Models[J]. arXiv preprint arXiv:2411.17002, 2024.

---

### Decision · Program_Chairs · 2026-04-30

**Decision:**

Accept (regular)

**Comment:**

This paper addresses online test-time adaptation (TTA) for multi-label classification using vision-language models (VLMs). While existing approaches largely ignore label co-occurrence, the proposed method incorporates such dependencies in a gradient-free manner by adjusting logits using a distribution over label correlations. Specifically, it selects high-confidence anchor labels and estimates co-occurrence via pointwise mutual information (PMI), enabling a closed-form, online correction without fine-tuning.

Reviewers highlighted several strengths of the paper. The work is clearly written with a well-motivated problem setting (tzvE, fmtz). The proposed approach is considered principled and well-founded, establishing a meaningful connection between test-time adaptation and Bayesian conditional priors, with an elegant formulation based on PMI that enables lightweight and closed-form calibration (tzvE, fmtz, PCds). In addition, the idea of incorporating label correlations into TTA is regarded as novel (PCds). The method is also computationally efficient, avoiding iterative optimization and enabling low-latency inference (tzvE, fmtz). Finally, extensive experiments demonstrate consistent and significant improvements over prior methods across multiple benchmarks (PCds, fmtz).

At the same time, reviewers raised concerns about the robustness of the anchor-based design, particularly its reliance on a single top-1 prediction, which may be unreliable and lead to error propagation without a clear mitigation mechanism (tzvE, fmtz). It is also unclear how this formulation affects long-tail classes, which may be suppressed due to limited co-occurrence statistics (tzvE). In addition, the method depends on the ordering and distribution of the test stream, potentially causing instability in early stages and sensitivity to non-i.i.d. conditions (PCds, fmtz). Finally, the evaluation is considered incomplete, with missing comparisons to recent CLIP-based TTA methods (fmtz).

Following the author response, the reviewers were generally satisfied with the clarifications and converged toward a positive assessment.

Some concerns still remain. The reliance on a single top-1 anchor continues to raise robustness issues: as reflected in Table 2, performance degradation can still occur, and there is currently no mechanism to determine whether the selected anchor is reliable, which may limit practical applicability. In addition, while the authors provided further analysis on long-tail performance, the evaluated datasets involve a relatively small number of classes (<100), making it unclear whether the observed behavior generalizes to more realistic large-scale long-tailed scenarios. Addressing these points more clearly would further strengthen the paper.